# Assigning harvested waterfowl to geographic origin using feather $\delta^2$H isoscapes: What is the best analytical approach?

**Jackson W. Kusack**[1,2]*, **Douglas C. Tozer**[2], **Kayla M. Harvey**[3], **Michael L. Schummer**[3], **Keith A. Hobson**[1,4]

**1** Department of Biology, Biological and Geological Sciences Building, Western University, London, Ontario, Canada, **2** Long Point Waterfowl and Wetlands Research Program, Birds Canada, Port Rowan, Ontario, Canada, **3** Department of Environmental Biology, State University of New York College of Environmental Science and Forestry, Syracuse, New York, United States of America, **4** Environment and Climate Change Canada, Saskatoon, Saskatchewan, Canada

* jkusack@uwo.ca

**Data Availability Statement:** All data files are available from the Dryad database (Kusack et al. [2023] Data from: Assigning harvested waterfowl to geographic origin using feather δ2H isoscapes:

## Abstract

Establishing links between breeding, stopover, and wintering sites for migratory species is important for their effective conservation and management. Isotopic assignment methods used to create these connections rely on the use of predictable, established relationships between the isotopic composition of environmental hydrogen and that of the non-exchangeable hydrogen in animal tissues, often in the form of a calibration equation relating feather ($\delta^2$H$_f$) values derived from known-origin individuals and amount-weighted long-term precipitation ($\delta^2$H$_p$) data. The efficacy of assigning waterfowl to moult origin using stable isotopes depends on the accuracy of these relationships and their statistical uncertainty. Most current calibrations for terrestrial species in North America are done using amount-weighted mean growing-season $\delta^2$H$_p$ values, but the calibration relationship is less clear for aquatic and semi-aquatic species. Our objective was to critically evaluate current methods used to calibrate $\delta^2$H$_p$ isoscapes to predicted $\delta^2$H$_f$ values for waterfowl. Specifically, we evaluated the strength of the relationships between $\delta^2$H$_p$ values from three commonly used isoscapes and known-origin $\delta^2$H$_f$ values three published datasets and one collected as part of this study, also grouping these data into foraging guilds (dabbling vs diving ducks). We then evaluated the performance of assignments using these calibrations by applying a cross-validation procedure. It remains unclear if any of the tested $\delta^2$H$_p$ isoscapes better predict surface water inputs into food webs for foraging waterfowl. We found only marginal differences in the performance of the tested known-origin datasets, where the combined foraging-guild-specific datasets showed lower assignment precision and model fit compared to data for individual species. We recommend the use of the more conservative combined foraging-guild-specific datasets to assign geographic origin for all dabbling duck species. Refining these relationships is important for improved waterfowl management and contributes to a better understanding of the limitations of assignment methods when using the isotope approach.

What is the best analytical approach? https://doi.org/10.5061/dryad.9w0vt4bmd).

**Funding:** This research was supported by funding from Birds Canada and the Long Point Waterfowl and Wetlands Research Program (https://www.birdscanada.org/) (DCT, JWK, KMH, MLS), Camp Fire Conservation Fund (https://www.campfirefund.org/) (KMH, MLS), Delta Waterfowl (https://deltawaterfowl.org/) (KMH, MLS), Ducks Unlimited (https://www.ducks.org/) (KMH, MLS), Environment and Climate Change Canada (https://www.canada.ca/en/environment-climate-change.html) (KAH), Long Island Wildfowl Heritage Group (KMH, MLS), NSERC (https://www.nserc-crsng.gc.ca/index_eng.asp) (JWK [PGS-D]; KAH [Discovery Grant 2017-04430]), Province of Ontario QEII-GST (https://osap.gov.on.ca/OSAPPortal/en/A-ZListofAid/PRDR019236.html) (JWK), SUNY College of Environmental Science and Forestry (https://www.esf.edu/) (KMH, MLS), Waterfowl Research Foundation (KMH, MLS), Western University (https://www.uwo.ca/) (JWK, KAH). Also with funding from the Black Duck Joint Venture, Bluff's Hunting Club, and SC Johnson. The funders had no role in study design, data collection and analysis, decision to publish, or preparation of the manuscript.

**Competing interests:** The authors have declared that no competing interests exist.

## Introduction

Establishing links between breeding, stopover, and wintering sites for migratory species is important for the effective conservation and management of those species and their habitats [1,2]. The development of extrinsic tracking tools has greatly increased our ability to establish patterns of migratory connectivity [3], but there are numerous situations involving unmarked individuals where only intrinsic markers are possible for inferring these connections. Such intrinsic markers typically involve the use of genetic or chemical molecular markers. This use of spatially explicit assignments to determine the origin of unmarked, migrant individuals using measurements of tissue stable-hydrogen isotopes ($\delta^2$H) has grown considerably over the past two decades (reviewed in [4]). In addition to numerous non-game animals, this isotopic approach has been applied to determine the geographic origins of several hunted waterfowl species across North America [5–10] and Eurasia [11–14]. In this context using the stable isotope approach, origin is generally not a specific location, but instead describes a probabilistically defined region that likely contains the location where an individual previously grew the sampled tissue(s) such as a natal, breeding, or non-breeding site. This assignment method has been important in improving our understanding of migratory connectivity, especially between breeding and harvest areas [8–10,15,16], and has the potential to contribute considerably to waterfowl management.

Isotopic assignment methods depend on the use of predictable relationships between the isotopic composition of environmental hydrogen (H) (e.g., precipitation, standing surface water) and non-exchangeable H in animal tissues. This approach relies on the fact that all H in animal tissues is derived ultimately from environmental H, either through diet or drinking water. Relationships between environmental H and non-metabolically active animal tissues formed locally, such as in feathers, allow inference of origins of individuals that are otherwise unmarked or tracked remotely, and so are not biased to focal (marked) populations within a species' range. Feather hydrogen-isotope values ($\delta^2$H$_f$) reflect the source of H in local food webs following isotopic discrimination (i.e., the preferential assimilation of the heavy ($^2$H) or light ($^1$H) form), which occurs during incorporation through food webs and ultimately into consumer tissues. Feathers are inert following formation, so the non-exchangeable $\delta^2$H values of feathers are representative of the environmental conditions present during feather growth [17], assuming no endogenous reserves are used during feather formation [18]. As waterfowl exhibit synchronous flight-feather moult after breeding [19], stable isotopes within these newly-formed feathers should reflect stable isotope abundance present within the environment at the moulting site for adults. Similarly, stable isotopes present in feathers from juvenile waterfowl should reflect stable isotopes present within the environment at the natal site of those individuals.

Relating these baseline environmental H isotope values, driven primarily by precipitation ($\delta^2$H$_p$), to $\delta^2$H$_f$ values requires a calibration or transfer function [20], often in the form of a linear equation (hereafter calibration equation). To derive these relationships, researchers primarily target known-origin, wild individuals whose tissues can confidently be related to a given location and relate their $\delta^2$H values to an averaged $\delta^2$H$_p$ at that location. For some animals, such as insects, calibrations can be derived in the laboratory using isotopically-known, dietary substrates [21,22]. Sample sizes and geographic coverage needed to adequately capture these broad-scale relationships often necessitates lumping of taxa with similar life history [17,23–29]. Over the past 30 years, these relationships have been derived for many taxa (see S1 Table), including bats [24,30–32], butterflies and moths [21,22,33], dragonflies [28], hoverflies [34], raptors [25,35–37], songbirds [23,38–41], and waterfowl (Table 1).

**Table 1. Summary of published calibration equations and associated statistics relating $\delta^2 H_p$ to $\delta^2 H_f$ for waterfowl, waterbirds, and shorebirds.**

| Common name | Latin | Calibration equation | $\delta^2 H_p{}^a$ | $r^2$ | $SD_{resid}$ | Source |
|---|---|---|---|---|---|---|
| **Anseriformes** | | | | | | |
| Lesser Scaup | *Aythya affinis* | $\delta^2 H_f = -31.6 + 0.93 * \delta^2 H_p$ | MGS$_{B-2005}$ | 0.78 | 12.8 ‰ | [42,43] |
| Mallard, Northern Pintail | *Anas platyrhynchos, Anas acuta* | $\delta^2 H_f = -57 + 0.83 * \delta^2 H_p$ | MGS$_M$ | - | - | [29] |
| - | - | $\delta^2 H_f = -61 + 0.67 * \delta^2 H_p$ | MA$_{B-2005}$ | - | - | [29] |
| Mallard | *Anas platyrhynchos* | $\delta^2 H_f = -21.9 + 1.36 * \delta^2 H_p$ | MGS$_{B-2005}$ | 0.61 | - | [44] |
| Swan Goose | *Anser cygnoides* | $\delta^2 H_f = 9.03 + 1.71 * \delta^2 H_p$ | MGS$_{B-2005}$ | 0.43 | 8.89 ‰ | [45] |
| **Waterbirds and Shorebirds** | | | | | | |
| Virginia Rail, King Rail | *Rallus limicola, R. elegans* | $\delta^2 H_f = -43.82 + 1.16 * \delta^2 H_p$ | MGS$_{B-2005}$ | 0.76 | 8.6 ‰ | [46] |
| 4 rail species | | $\delta^2 H_f = -74 + 1.16 * \delta^2 H_p$ | MM$_B{}^b$ | - | - | [47] |
| 14 wader species | | $\delta^2 H_f = -37.56 + 0.34 * \delta^2 H_p$ | MGS$_B{}^b$ | - | - | [27] |

$SD_{resid}$, Standard deviation of residuals.

[a] MGS$_{B-2005}$, *Amount-weighted mean growing-season precipitation $\delta^2 H$* [38]; MGS$_M$, [48]; MA$_{B-2005}$, *Amount-weighted mean annual precipitation $\delta^2 H$* [38,49]; MM$_B$, *Amount-weighted mean monthly precipitation (Nov-Feb; Apr-Aug) $\delta^2 H$* [50]; MGS$_B$, [50].

[b] Isoscape was downloaded at the time of publication and may not represent the current form available in the reference.

'-' indicates information repeated from the line above; blanks indicate unreported information.

Calibration equations estimating the transfer of H from precipitation to tissue can vary among taxa and age classes within taxa [44,51], as life history and foraging strategies influence isotopic source and routing [23]. For example, for individual songbirds captured at the same moulting location, species is an important predictor of $\delta^2 H_f$ values [52]. Waterfowl species can be broadly grouped into two guilds with differing foraging strategies: dabblers (i.e., feed on aquatic vegetation and invertebrates beneath the surface of the water) and divers (i.e., dive to feed upon fish, invertebrates, and vegetation). Although the diets and behaviour of dabbling and diving ducks are varied and can overlap, these broad foraging strategies partition the dietary niche of these ducks to different microhabitats (dabblers–surface; divers–benthos or water column), which could theoretically influence these calibration relationships, although this is largely unknown for waterfowl.

When utilizing precipitation isoscape to assign individuals to origin, it is important to understand how H in precipitation contributes to H in a consumer's tissues. For terrestrial-foraging species, most current calibrations in North America are done using amount-weighted mean growing-season (hereafter MGS) $\delta^2 H_p$ isoscapes, which incorporate isotope data for months with average temperatures > 0°C [38,48] (S1 Table). These calibrations work on the assumption that consumer $\delta^2 H_f$ will relate to the $\delta^2 H_p$ during the period of greatest vegetative growth, as these precipitation signals are translated into plant biomass and to consumers. The appropriate calibration is less clear for aquatic and semi-aquatic species or those that eat foods that occur in aquatic emergent plant communities. Despite this, the focus for waterfowl calibration studies has been on the relationship between consumer tissues and MGS $\delta^2 H_p$ values, as all but one waterfowl calibration relationship has utilized MGS $\delta^2 H_p$ values (Table 1), although few studies have directly measured surface water $\delta^2 H$ to compare with consumer tissues [53,54]. The other isoscape used is the amount-weighted mean annual (hereafter MA) $\delta^2 H_p$ grid, which incorporates precipitation isotope data across all months [38,55,56]. The main difference for the MA grid is the potential contribution of snowmelt to the surface water. Although no studies have directly measured this relationship, snowmelt entering waterbodies likely influences dietary $\delta^2 H$ especially for northern moulting waterfowl. Therefore, it is not clear which isoscape captures this relationship more accurately.

The efficacy of assigning waterfowl to a geographic origin using the stable isotope approach also depends upon the accuracy of the calibration relationship between $\delta^2H_f$ and $\delta^2H_p$, as well as the variance one might expect for such a calibration. This involves isotopic measurement error [57] and intrinsic differences between individuals (e.g., behaviour, metabolism), in addition to error associated with the derivation of $\delta^2H_p$ isoscapes (i.e., $\delta^2H_p$ measurement error, interpolation uncertainty, annual environmental effects). As such, modern $\delta^2H_p$ isoscape grids are generally accompanied by a spatially-explicit estimate of $\delta^2H_p$ variability [50], where error is generally greater in regions with fewer sampling points [49]. To capture the remaining calibration error, variability is often approximated using the standard deviation of calibration model residuals (hereafter $SD_{resid}$), which includes uncertainty in $\delta^2H_f$ values (e.g., measurement error, inter- and intraspecific intrinsic variability) and site-specific $\delta^2H_p$ values (e.g., interpolation error, climatic variability). For waterfowl, annual climatic variation such as dry summers leading to increased evaporation in shallow ponds and more positive surface water $\delta^2H$ values [53] likely contributes to increased variability. Propagating as much of the known error as possible into assignments is the objective of most practitioners and with the adoption of newer assignment algorithms [58,59] these sources of error can be incorporated into likelihood-based assignment algorithms, to provide the most complete estimates of assignment error.

The primary goal of our research was to critically evaluate current methods used to calibrate precipitation-hydrogen isoscapes to predicted $\delta^2H_f$ values and, by extension, evaluate likelihood-based assignment methods for waterfowl. Specifically, we aimed to test whether known-origin waterfowl $\delta^2H_f$ values correlated better with MA $\delta^2H_p$ or with MGS $\delta^2H_p$, and which calibration relationship is best for different foraging guilds of ducks (dabbling vs. diving). To test these correlations, we used published $\delta^2H_f$ data for known-origin waterfowl and collected additional data from across northeastern North America, a region that has been unrepresented to date. Using these data and published isoscapes, we derived calibrations between measured and predicted $\delta^2H_f$ values and then evaluated the accuracy and precision of assignments by applying a cross-validation procedure. Lastly, as a proof-of-concept, we reanalyzed a published dataset [9], applied our derived calibration methods, and compared the results to the previous utilized method.

## Materials and methods

### Isoscapes

We compiled three $\delta^2H_p$ isoscapes from two sources that represent the most complete precipitation isoscapes available at the time of publication. Both sources utilized the long-term datasets compiled by the Global Network of Isotopes in Precipitation (GNIP) of the International Atomic Energy Association (IAEA) [60]. From the WaterIsotopes website [50], we obtained a predicted amount-weighted (i.e., weighted by the monthly amount of precipitation) mean annual $\delta^2H_p$ grid (5 arc-minute resolution, hereafter $MA_B$), amount-weighted mean growing-season $\delta^2H_p$ grid (5 arc-minute resolution, hereafter $MGS_B$), and associated uncertainty grids (1 standard deviation) for $MGS_B$ and $MA_B$ predictions. Using monthly station-specific $\delta^2H_p$ values (largely from GNIP), these isoscapes are typically interpolated using algorithms that rely on spatial (e.g., latitude, elevation) correlates to account for variation in $\delta^2H_p$ [38,49], although the versions we used include more recent precipitation $\delta^2H_p$ data [61]. From the IAEA [60], we obtained an amount-weighted mean annual $\delta^2H_p$ grid (30 arc-second resolution, hereafter $MA_T$; accessed August 23, 2021), modelled using the updated Regionalized Cluster-Based Water Isotope Prediction Version 2 model (RCWIP2) [56]. This model updated the previous RCWIP model [55] and included an additional 7 years of $\delta^2H_p$ data (1960–2006).

In addition to the spatial correlates included in the [38,49] model (i.e., latitude and altitude), the RCWIP2 model included additional climatic (e.g., air temperature, vapour pressure) and geographical predictors (e.g., land mass fraction; for a complete list see [56]). The RCWIP2 isoscape grids are available as 1800 x 1800 arcminute GeoTIFF files (accessed May 27, 2022), which we downloaded separately and combined into a global grid [56]. These RCWIP2 isoscapes provide $\delta^2H_p$ values at the highest resolution available, but this resolution was not logistically feasible because of computer processing time. Therefore, we reduced the resolution to match the 5 arc-minute resolution of the $MGS_B$ and $MA_B$ isoscapes (method: bilinear resampling).

## Samples

We collected feathers from known-origin waterfowl across eastern Canada and the United States (n = 273, 2017–2021, hereafter the 'Kusack' dataset; see Table 2 for sample sizes by province and state and Fig 1 for geographic distribution). Most feather samples were collected from flightless hatch-year (HY) birds (i.e., 'locals') during regular banding operations, where feathers were collected opportunistically or during targeted sampling. We focused collection on

**Table 2. Summary statistics for feather stable-hydrogen isotope ($\delta^2H_f$) values.**

| Country | Province | Species | n | Foraging Guild | Mean ± SD $\delta^2H_f$ (‰)[a] |
|---|---|---|---|---|---|
| Canada | NB | American Black Duck | 3 | Dabbling | -100.3 ± 1.3 |
| | NS | American Black Duck | 1 | Dabbling | -107.5 |
| | ON | American Black Duck | 3 | Dabbling | -97.3 ± 5.7 |
| | | Green-winged Teal | 5 | Dabbling | -105.5 ± 16.0 |
| | | Blue-winged Teal | 2 | Dabbling | -110.4 ± 1.8 |
| | | Common merganser | 1 | Diving | -112.3 |
| | | Mallard | 7 | Dabbling | -118.4 ± 12.8 |
| | | Ring-necked Duck | 9 | Diving | -132.6 ± 16.7 |
| | | Wood Duck | 12 | Dabbling | -115.6 ± 21.0 |
| | QC | American Black Duck | 20 | Dabbling | -120.0 ± 8.0 |
| | | Mallard | 8 | Dabbling | -130.2 ± 7.3 |
| USA | CT | Mallard | 1 | Dabbling | -111.9 |
| | IN | Mallard | 1 | Dabbling | -76.3 |
| | MA | Mallard | 10 | Dabbling | -100.2 ± 4.3 |
| | | Wood Duck | 6 | Dabbling | -97.9 ± 13.2 |
| | MD | Wood Duck | 22 | Dabbling | -94.6 ± 14.8 |
| | MI | Mallard | 2 | Dabbling | -90.03 ± 20.3 |
| | NJ | Mallard | 1 | Dabbling | -82.5 |
| | NY | Mallard | 4 | Dabbling | -123.6 ± 5.8 |
| | | Wood Duck | 5 | Dabbling | -104.1 ± 15.1 |
| | OH | Wood Duck | 3 | Dabbling | -108.2 ± 8.7 |
| | PA | Mallard | 2 | Dabbling | -111.0 ± 9.4 |
| | | Wood Duck | 3 | Dabbling | -108.4 ± 14.6 |
| | VA | Wood Duck | 9 | Dabbling | -88.5 ± 9.8 |
| | WI | Mallard | 88 | Dabbling | -119.8 ± 16.4 |
| | | Ring-necked Duck | 23 | Diving | -119.9 ± 14.1 |
| | | Wood Duck | 24 | Dabbling | -105.3 ± 7.6 |
| Total | | | 275 | | |

[a] Mean $\delta^2H_f$ values were calculated without the outlier samples and sites (see Results).

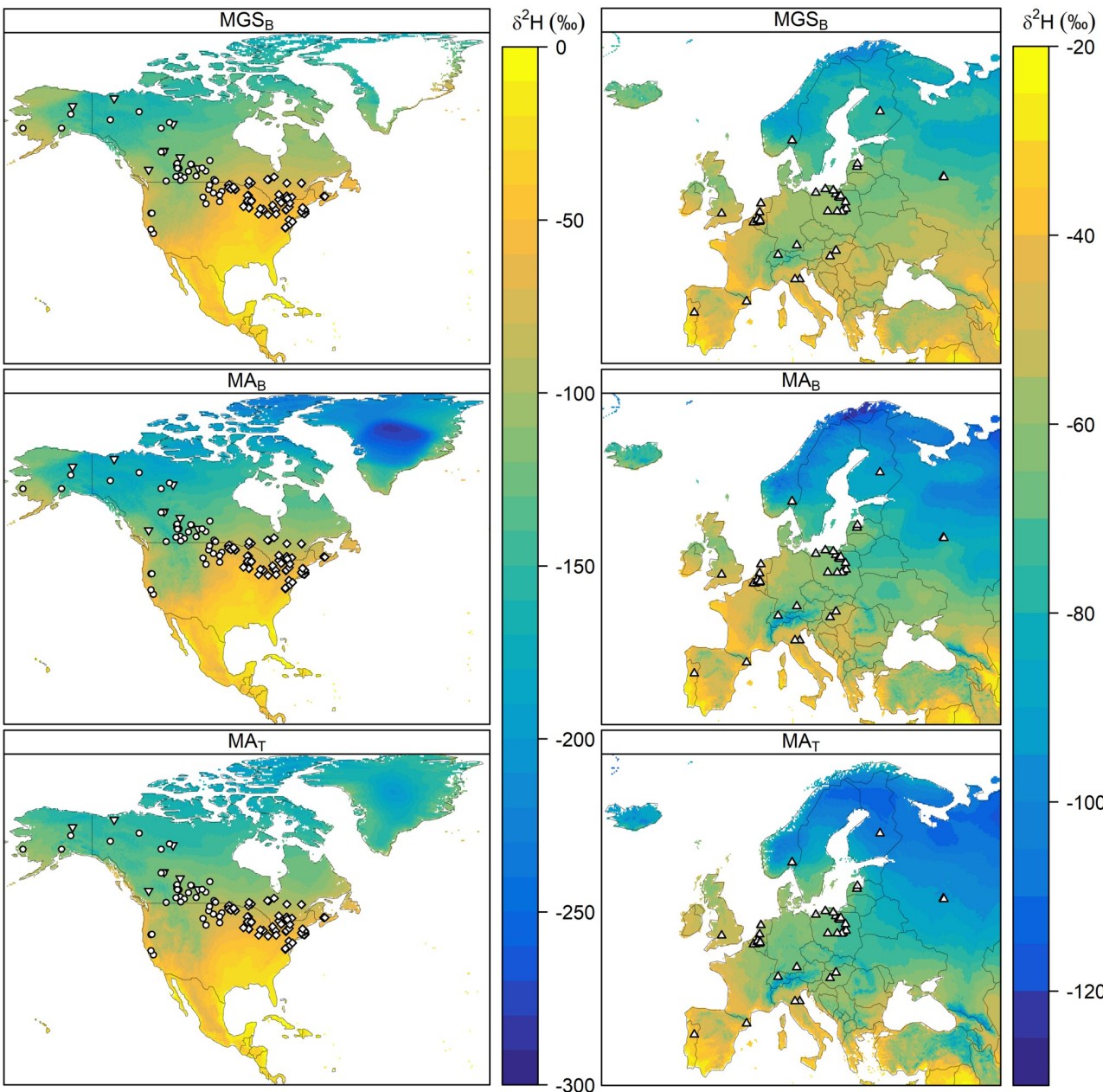

**Fig 1. Map of sampling sites.** Collection locations for North American and European known-origin ducks, overlayed on amount-weighted mean growing-season precipitation (MGS_B) [50] and amount-weighted mean annual precipitation (MA_B, [50]; MA_T, [56]) isoscape grids. Points show sampling location for individual samples and symbology shows the source publication (*triangle*, n = 212, van Dijk [44]; *circle*, n = 324, Hebert [29]; *diamond*, n = 275, Kusack; *inverted triangle*, n = 75, Clark [42,43]). Stable-hydrogen isotope values within each column are represented by the colour (scale is consistent among isoscape sources within the same continent). For more specific sampling locations for a given dataset, see the original publications.

primary (P1; clip ¼ inch of the distal end of the feather) and covert (secondary covert; pluck entire feather) feathers, but due to the opportunistic nature of sampling and the different ages at which banding occurred for HY birds, multiple different feather groups, including breast feathers (n = 17) were included in analyses. We sampled HY American Black Duck *Anas*

*rubripes*, Mallard *Anas platyrhynchos*, Ring-necked Duck *Aythya collaris*, and Wood Duck *Aix sponsa*. We also obtained Blue-winged Teal *Spatula discors* (n = 2) samples that were collected by [9]. Moulting adults (n = 9) were included if they were not flight-capable yet, but only newly moulted primary feathers were sampled from these birds to be sure of the local signal. Outside of banding stations, we also obtained primary feather tissue (P1) from HY Wood Ducks (n = 22) banded during a Maryland breeding study, which were sampled within 5 km of their original banding site as flightless young. We obtained flight feathers (primary [P1] and primary coverts) from the Species Composition Survey [62] when wings from HY or adults in incomplete moult were submitted from known origins (Green-winged Teal *Anas crecca* n = 5, American Black Duck n = 1, Ring-necked Duck = 3, Common Merganser *Mergus merganser* n = 1, Wood Duck n = 2).

We obtained published known-origin $\delta^2H_f$ data from the assignR known-origin dataset repository [58] and authors directly. For dabbling ducks, we obtained $\delta^2H_f$ data on known-origin Mallard and Northern Pintail pre-fledged HY birds captured in western North America [29] (n = 324, hereafter the 'Hebert' dataset) and known-origin juvenile and moulting adult Mallard across Europe [44] (n = 215, hereafter the 'van Dijk' dataset). Three samples from the van Dijk dataset were excluded from analyses as they did not overlap with the $MA_T$ isoscape (IDs 2755, 2932, and 2933). For diving ducks, we obtained data on known-origin HY Lesser Scaup *Aythya affinis* in western North America [42,43] (n = 75, hereafter the 'Clark' dataset).

## Stable isotope measurements

Feather samples were processed for $\delta^2H_f$ measurement at the Laboratory for Stable-isotope Science—Advanced Facility for Avian Research (n = 71; LSIS-AFAR; Western University, London, ON, CA) and the Cornell Isotope Laboratory (n = 204; COIL; Cornel, Ithaca, NY, USA). Feathers were first cleaned of surface oils by soaking and rinsing in a 2:1 chloroform: methanol mixture and allowed to dry under a fume hood. We sampled the distal end of the feather vane and weighed 0.350 ± 0.03 mg of feather material into silver capsules. At LSIS-A-FAR crushed capsules were then placed in a Uni-Prep carousel (Eurovector, Milan, Italy) heated to 60˚C, evacuated and then held under positive He pressure. Feather samples were combusted using flash pyrolysis (~1350˚C) on glassy carbon in a Eurovector elemental analyzer (Eurovector, Milan, Italy) coupled with a Thermo Delta V Plus continuous-flow isotope-ratio mass spectrometer (CF-IRMS; Thermo Instruments, Bremen, Germany). At COIL, the same procedures were followed, except feather samples were combusted ($>$ 1400˚C) using a Thermo Scientific Temperature Conversion Elemental Analyzer coupled via a Conflo IV (Thermo Scientific) to a Thermo Scientific Delta V CF-IRMS. Both labs used the comparative equilibration method of [63] using the same two keratin standards (CBS, $\delta^2H$ = -197 ‰; KHS, $\delta^2H$ = -54.1 ‰) corrected for linear instrumental drift. All results are reported for non-exchangeable H expressed in the typical delta notation, in units of per mil (‰), and normalized on the Vienna Standard Mean Ocean Water (VSMOW) scale. Based on within-run (n = 5 CBS at LSIS-AFAR; n = 7–9 Keratin at COIL) and across-run (n = 10 at LSIS-AFAR; n = 13 at COIL) analyses of standards, measurement error was approximately ± 2.5 ‰ (LSIS-AFAR) and ± 2.2 ‰ (COIL). All $\delta^2H_f$ values are reported relative to the Vienna Standard Mean Ocean Water–Standard Light Antarctic Precipitation scale. All published data used in our study were processed using the same comparative equilibration methods [63], using the same standards as [63] (i.e., CFS, CHS, BWB) or used standards that have been calibrated relative to the standards in [63] (i.e., KHS, CBS), and therefore should be comparable without any additional transformations [64].

## Statistics

All statistics were performed within the R statistical environment [65] (v. 4.2.2) using RStudio [66] (v. 2022.12.0). Spatial data manipulations were performed using the packages 'sf' [67] (v. 1.0–9) and 'terra' [68] (v. 1.6–47). All isoscape depictions and assignment procedures were done using the original coordinate system of the isoscapes (WGS84; EPSG:4326), but final depictions of assignment maps were converted to an Albers equal-area projection for North America (NAD83; EPSG:42303).

We used general linear models to derive calibration equations based on the relationship between known-origin $\delta^2H_f$ values and $\delta^2H_p$ values at the location of sampling. We removed outliers on a site-specific basis, where individuals with $\delta^2H_f$ values more positive than the third quartile + 1.5 x the interquartile range, for that site, were removed from the calibration, as were those with $\delta^2H_f$ values more negative than the first quartile– 1.5 x the interquartile range. Separate calibration equations were derived for each published known-origin data source [29,42–44], as well as our data, paired with each precipitation isoscape [50,56]. We also grouped all dabbling ducks (American Black Duck, Blue-winged Teal, Green-winged Teal, Mallard, Wood Duck; hereafter the 'Dabblers' dataset) and diving ducks (Common Merganser, Lesser Scaup, Ring-necked Duck; hereafter the 'Divers' dataset). From each calibration equation, we reported the $SD_{resid}$ and adjusted $r^2$ to approximate model fit.

## Model validation

To validate the performance of known-origin $\delta^2H_f$ datasets, isoscapes, and any resulting calibration ($\delta^2H_f$ vs $\delta^2H_p$) equations, we performed a cross-validation procedure, similar to those used by Ma et al. [58]. Half of the given dataset, chosen at random, was used to produce a calibration equation, while the other half was used to validate the derived equation. These isoscapes were then converted to predicted $\delta^2H_f$ isoscapes using the calibration equation derived from the calibration subset. As the known-origin data were collected within North America and Europe, isoscapes were limited to these continents. Specifically, North America (extent: longitude -170 to -10˚, latitude 7 to 84˚) was masked to exclude South America while Europe (extent: longitude -25 to 50˚, latitude 35 to 72˚) was masked to exclude Africa. Geopolitical shapefiles were obtained from the R package 'rnaturalearth' [69] (v. 0.1.0). Isoscapes were not masked to any breeding range since we were assessing multispecies data, and waterfowl can migrate to moult outside of the breeding range [70].

We then assessed the likelihood that any given cell within the $\delta^2H_f$ isoscape was the origin of an individual duck using the procedures and functions from the isocat package (v. 0.2.6 [59]), which uses normal probability density function [20,71] incorporating both calibration error ($\sigma_{calibration}$) and isoscape error ($\sigma_{isoscape}$) into the expected standard deviation of a given isoscape cell ($\sigma_c$). Calibration error ($SD_{resid}$) was derived directly from the residuals of the calibration relationship between the isoscape and calibration data. Isoscape error was extracted directly from the isoscape uncertainty raster. For the $MA_T$ isoscape, which did not have an error grid, we used a placeholder grid with no uncertainty (i.e., $\sigma_{isoscape} = 0$), which simplified the error calculation to just the calibration error, while still using the isocat functions. Probabilities were normalized to sum to 1, estimating a probability of origin, and the upper 66.6% of probabilities of origin (i.e., a 2:1 odds ratio) were selected, creating a uniform region of likely origins (i.e., all cells are equally likely). As some of the grouped datasets contained samples from both North America and Europe, we limited the likely origins for a given individual to the continent (see above) where sampled. We evaluated the performance of each known-origin dataset and isoscape using estimates of accuracy, precision, and minimum distance. We measured accuracy by determining the proportion of individuals that were correctly assigned

under the applied odds ratio (i.e., the binary grid contains the sampling point for known-origin feathers, [72]). Other validation methods examined the performance of these thresholds on a spectrum (0–1) [58,59], rather than a single odds ratio, but as we were focussing on the calibration data and isoscapes rather than the assignment methods, we chose to examine the performance of these methods using a single, conservative, odds ratio instead (2:1). We measured precision as the proportion of cells in the raster which were likely assigned compared to the total number of cells [72]. This procedure was repeated 25 times for each dataset and isoscape pairing, with precision and minimum distance being summarized as the mean value across individuals within a given iteration. For inaccurately assigned individuals, we also measured the minimum distance (km) between the location of sampling for the known-origin individual and the closest cell of the region of likely origin (hereafter 'minimum distance'). We followed methods from [59] but used the function 'distance' within the package 'terra'.

### Test dataset

As a proof-of-concept, we reanalyzed data from [9] on Blue-winged Teal harvested across Canada (2014–2018; n = 144). This study represents a case where a diving duck (Lesser Scaup) calibration [42,43] was utilized to assign the geographic origins of a dabbling duck. This has been common in waterfowl studies to date, as six of eight published assignments for unknown-origin dabbling ducks or geese have used this equation [6–9,12,13,73,74]. We chose to reanalyze [9], as this study has direct management implications for the connectivity of harvested Blue-winged Teal. To facilitate direct comparison to the original publication, as these data were assigned separately for each harvest region, we subsetted these data and only analyzed birds harvested in the southern Saskatchewan harvest region because of its larger sample size (n = 47).

To assess the consequences of using different calibration equations to assign waterfowl origins, we repeated the assignment procedures above, but with the updated $\delta^2H_p$ grids [38,50,56] and the calibration equations derived from the Dabblers and Divers datasets. Therefore, we applied six assignments: $MGS_B$ ~ Dabblers, $MA_B$ ~ Dabblers, $MA_T$ ~ Dabblers, $MGS_B$ ~ Divers, $MA_B$ ~ Divers, and $MA_T$ ~ Divers. We masked this isoscape to the Blue-winged Teal breeding range [75]. As these are unknown origin samples, there is no way to truly validate the accuracy of any method, but our intention was simply to demonstrate the scale of differences relative to each other.

## Results

We collected and processed $\delta^2H_f$ values of 273 known-origin ducks (American Black Duck [n = 27], Green-winged Teal [n = 5], Blue-winged Teal [n = 2], Common Merganser [n = 1], Mallard [n = 124], Ring-necked Duck [n = 32], Wood Duck [n = 82]) across eastern North America (2017–2021; Table 2). One site in Wisconsin showed more positive $\delta^2H_f$ values than expected (mean (sd) = –85.8 ‰ (10.3 ‰); Mallard [n = 7] and Wood Duck [n = 4]) likely due to irrigation and increased effect of evapotranspiration in shallow water due to the site's location on a cranberry farm. We removed 35 outliers whose $\delta^2H_f$ values deviated from the site-specific mean $\delta^2H_f$ (Clark [n = 2], van Dijk [n = 10], Hebert [n = 19], Kusack [n = 4]). These samples were excluded from all further analyses.

### Calibration equations

Calibration relationships were generally consistent, with a positive linear relationship between $\delta^2H_f$ and $\delta^2H_p$ values within each dataset (Figs 2 and 3). All calibration equations had a negative intercept term (range: –82.6 to –9.9 ‰) and a positive slope term (range: 0.5 to 1.2), but

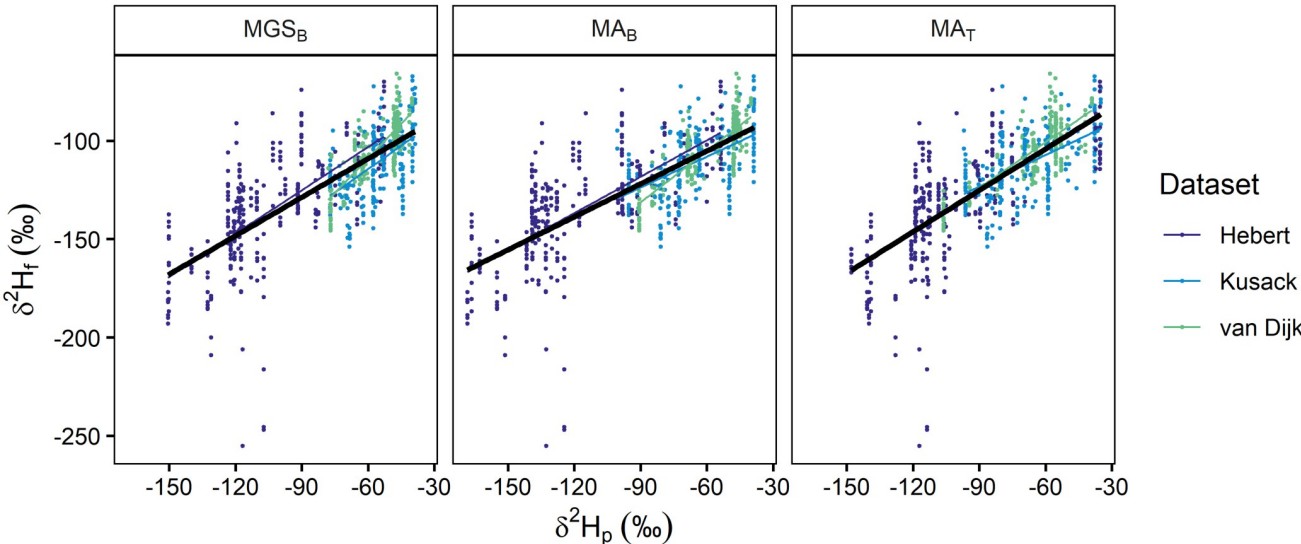

**Fig 2. Dabbling duck calibration relationships.** Linear relationships between $\delta^2H_p$ (amount-weighted mean growing-season precipitation, MGS$_B$ [50]; amount-weighted mean annual precipitation, MA$_B$ [50]; MA$_T$ [56]) and $\delta^2H_f$ from known-origin dabbling ducks. The solid black line shows an overall linear relationship and smaller lines show dataset-specific calibration relationships. Points show individual known-origin ducks, separated by dataset (shown in different colours).

the magnitude of these terms varied (Table 3). Within each respective known-origin dataset, calibration equation model fit was only marginally different when derived using $\delta^2H_p$ values that were extracted from the three different isoscapes (Table 3). Comparing different known-origin datasets there was a greater difference in model fit. For dabbling ducks (Fig 2), the calibration equation derived using the Hebert dataset showed almost double the amount of residual variation (~21 ‰) compared to van Dijk (~10 ‰) and Kusack (~14 ‰). For the

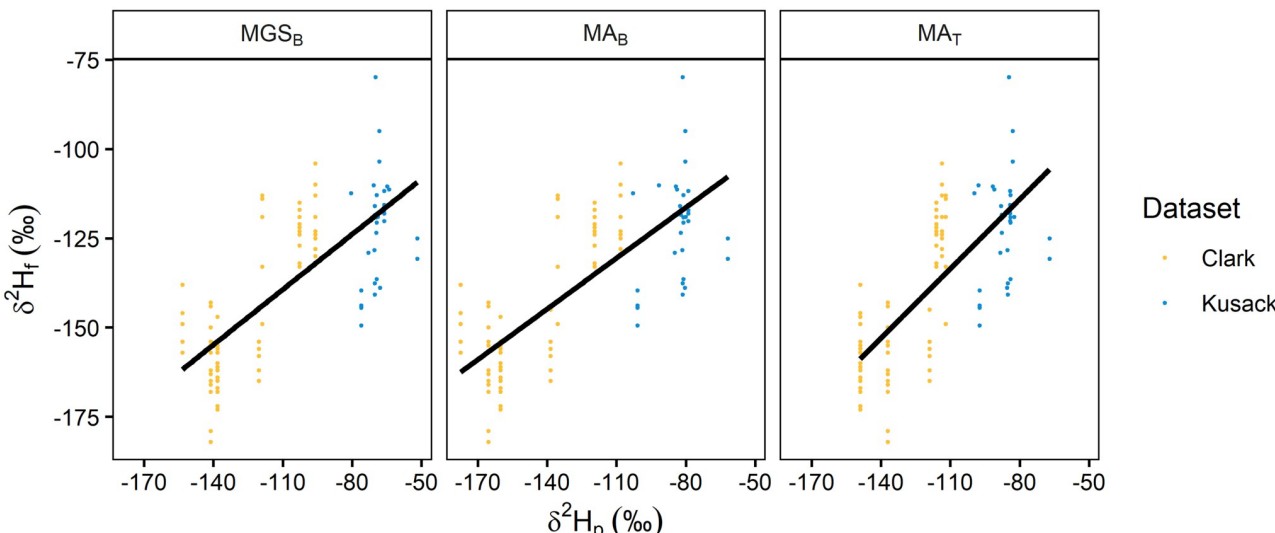

**Fig 3. Diving duck calibration relationships.** Linear relationships between $\delta^2H_p$ (amount-weighted mean growing-season precipitation, MGS$_B$ [50]; amount-weighted mean annual precipitation, MA$_B$ [50]; MA$_T$ [56]) and $\delta^2H_f$ from known-origin diving ducks. The solid black line shows an overall linear relationship. Points show individual known-origin ducks, separated by dataset (shown in different colours).

**Table 3. Summary of derived calibration equations.**

| Source | Isoscape[a] | Calibration | n | $SD_{resid}$ | $r^2$ |
|---|---|---|---|---|---|
| **Dabblers** | $MGS_B$ | $\delta^2H_f = -69.9 + 0.7 * \delta^2H_p$ | 734 | 17.7 | 0.56 |
| | $MA_B$ | $\delta^2H_f = -71.7 + 0.6 * \delta^2H_p$ | 734 | 17.2 | 0.58 |
| | $MA_T$ | $\delta^2H_f = -62.0 + 0.7 * \delta^2H_p$ | 734 | 17.3 | 0.58 |
| van Dijk [44] | $MGS_B$ | $\delta^2H_f = -38.5 + 1.2 * \delta^2H_p$ | 202 | 10.0 | 0.65 |
| | $MA_B$ | $\delta^2H_f = -52.9 + 0.9 * \delta^2H_p$ | 202 | 9.6 | 0.68 |
| | $MA_T$ | $\delta^2H_f = -54.2 + 0.8 * \delta^2H_p$ | 202 | 9.9 | 0.65 |
| Hebert [29] | $MGS_B$ | $\delta^2H_f = -59.1 + 0.7 * \delta^2H_p$ | 305 | 21.7 | 0.44 |
| | $MA_B$ | $\delta^2H_f = -62.7 + 0.6 * \delta^2H_p$ | 305 | 21.6 | 0.45 |
| | $MA_T$ | $\delta^2H_f = -62.9 + 0.7 * \delta^2H_p$ | 305 | 22.0 | 0.43 |
| Kusack (Dabblers) | $MGS_B$ | $\delta^2H_f = -66.9 + 0.8 * \delta^2H_p$ | 227 | 14.8 | 0.28 |
| | $MA_B$ | $\delta^2H_f = -76.7 + 0.5 * \delta^2H_p$ | 227 | 14.9 | 0.27 |
| | $MA_T$ | $\delta^2H_f = -75.1 + 0.5 * \delta^2H_p$ | 227 | 14.4 | 0.32 |
| **Divers** | $MGS_B$ | $\delta^2H_f = -82.6 + 0.5 * \delta^2H_p$ | 106 | 14.7 | 0.54 |
| | $MA_B$ | $\delta^2H_f = -78.4 + 0.5 * \delta^2H_p$ | 106 | 14.1 | 0.58 |
| | $MA_T$ | $\delta^2H_f = -63.1 + 0.6 * \delta^2H_p$ | 106 | 14.8 | 0.54 |
| Clark [42,43] | $MGS_B$ | $\delta^2H_f = -37.5 + 0.9 * \delta^2H_p$ | 73 | 12.6 | 0.63 |
| | $MA_B$ | $\delta^2H_f = -41.7 + 0.7 * \delta^2H_p$ | 73 | 12.4 | 0.64 |
| | $MA_T$ | $\delta^2H_f = -9.9 + 1.0 * \delta^2H_p$ | 73 | 13.1 | 0.60 |

$SD_{resid}$, standard deviation of residuals.

[a] $MGS_B$, Amount-weighted mean growing-season precipitation [50]; $MA_B$, Amount-weighted mean annual precipitation [50]; $MA_T$, Amount-weighted mean annual precipitation [56].

calibration derived from the Dabblers dataset, model fit improved marginally (~17 ‰) compared to Hebert but was still greater than the other individual datasets. For diving ducks (Fig 3), where only a few additional samples (n = 33) were added to the Clark dataset, model fit was reduced slightly (Table 3).

## Model validation

Accuracy in assignment did not differ consistently among isoscapes when considering the same known-origin dataset but differed marginally among known-origin datasets (Fig 4). Estimates of mean accuracy from cross-validation procedures applied to dabbling duck datasets (van Dijk, Hebert, Kusack, Dabblers) all fell within a proportion of 0.66 ± 0.07 accurately assigned individuals (range = 0.63 to 0.73) while estimates for diving duck datasets (Clark, Divers) all fell within 0.66 ± 0.05 (range = 0.61 to 0.71), both consistent with the accuracy that we expect for our applied odds ratio (i.e., 0.66 for 2:1). The Dabblers dataset showed the highest accuracy, which was greater than expected (> 0.66), in all but four iterations across all three isoscapes. For the diving duck datasets, accuracy was more variable, and consistently lower than expected, on average, for the Divers dataset (accuracy < 0.66 for 61 of 75 iterations). Minimum distance showed some variability between datasets, but no strong differences were identified between the three isoscapes (Fig 4). Specifically, the van Dijk and Clark datasets showed the lowest values for minimum distance, but otherwise, the datasets showed similar minimum distances (mean (sd) 290 km (52) Dabblers; 298 km (63) Hebert; 246 km (45) Kusack).

Mean precision was more variable between datasets compared to accuracy (Fig 4). For the van Dijk, Hebert, Clark, and Divers datasets, the $MA_B$ calibrations showed the best precision

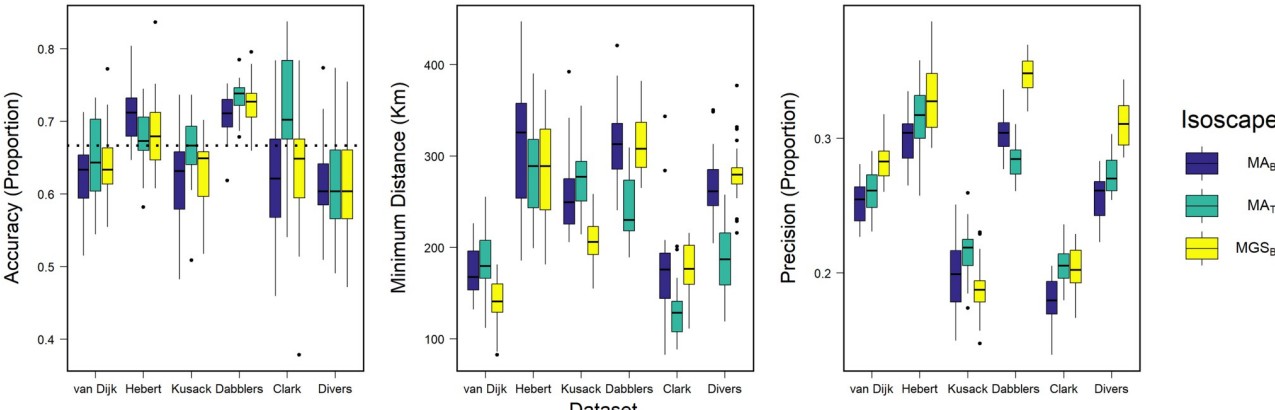

**Fig 4. Accuracy, precision, and minimum distance distributions.** Box and whisker plots showing distributions of accuracy, minimum distance, and precision from cross-validation procedures applied to the listed datasets. Validation results for different isoscapes (amount-weighted mean growing-season precipitation, MGS$_B$ [50]; amount-weighted mean annual precipitation, MA$_B$ [50]; MA$_T$ [56]) are shown in different colours. Boxplots show medians (horizontal lines within boxes), 25th and 75th quantiles (upper and lower limits of the boxes), upper and lower extreme values (whiskers), and outliers as values outside of 1.5 x the interquartile range (points). Dotted line (left) shows the expected accuracy for the applied odds ratio (2:1 or 0.66).

follow by the MA$_T$ and MGS$_B$ grids respectfully, but again showed no consistent differences among the isoscapes for the Kusack and Dabblers datasets (Fig 4). The Kusack and Clark datasets showed the best precision (Fig 4), despite the Kusack dataset having the lowest r$^2$ (Table 3). For the Dabblers and Hebert datasets, precision was low and did not exceed 0.4 (Fig 4). For the Divers dataset, precision and accuracy were lower with the inclusion of the diving ducks from the Kusack dataset (Fig 4), despite a slight increase in sample size.

## Test dataset

Likely origins were not noticeably altered by using the updated calibration equations for Dabblers or Divers (Fig 5), which showed likely origins of Blue-winged Teal in the northwestern Boreal Forest of northern British Columbia and Alberta (see S1 Fig for a recreation of the figure from [9]). This general result matches the results in the original publication, although at a higher resolution. Using the calibration equation derived from the Divers dataset tended to bias likely origins towards the northwest compared to using the Dabblers calibration, which showed similar origins to the original publication apart from greater likelihood to the south (Fig 5). As the residual standard deviation was greater (range = 14.1–17.7 ‰) than what was used in the original publication (12.8 ‰), the larger area of potential origin is not unexpected. The original publication also did not include isoscape uncertainty, as we have included in the MGS$_B$ and MA$_B$ assignments, which likely contributes to the broader areas. Aside from minor fluctuations in the upper and lower range and the maximum number of individuals assigned, the use of any specific isoscape did not significantly alter the final depiction, within each foraging guild.

## Discussion

Combining published and newly collected data, we 1) critically evaluated the relationship between $\delta^2$H values of waterfowl feathers and precipitation at known sites of feather growth; 2) empirically tested the performance of three publicly-available isoscapes and four calibration datasets, including our own samples; and 3) derived updated calibration equations for diving and dabbling ducks that can be applied to future waterfowl studies in North America and

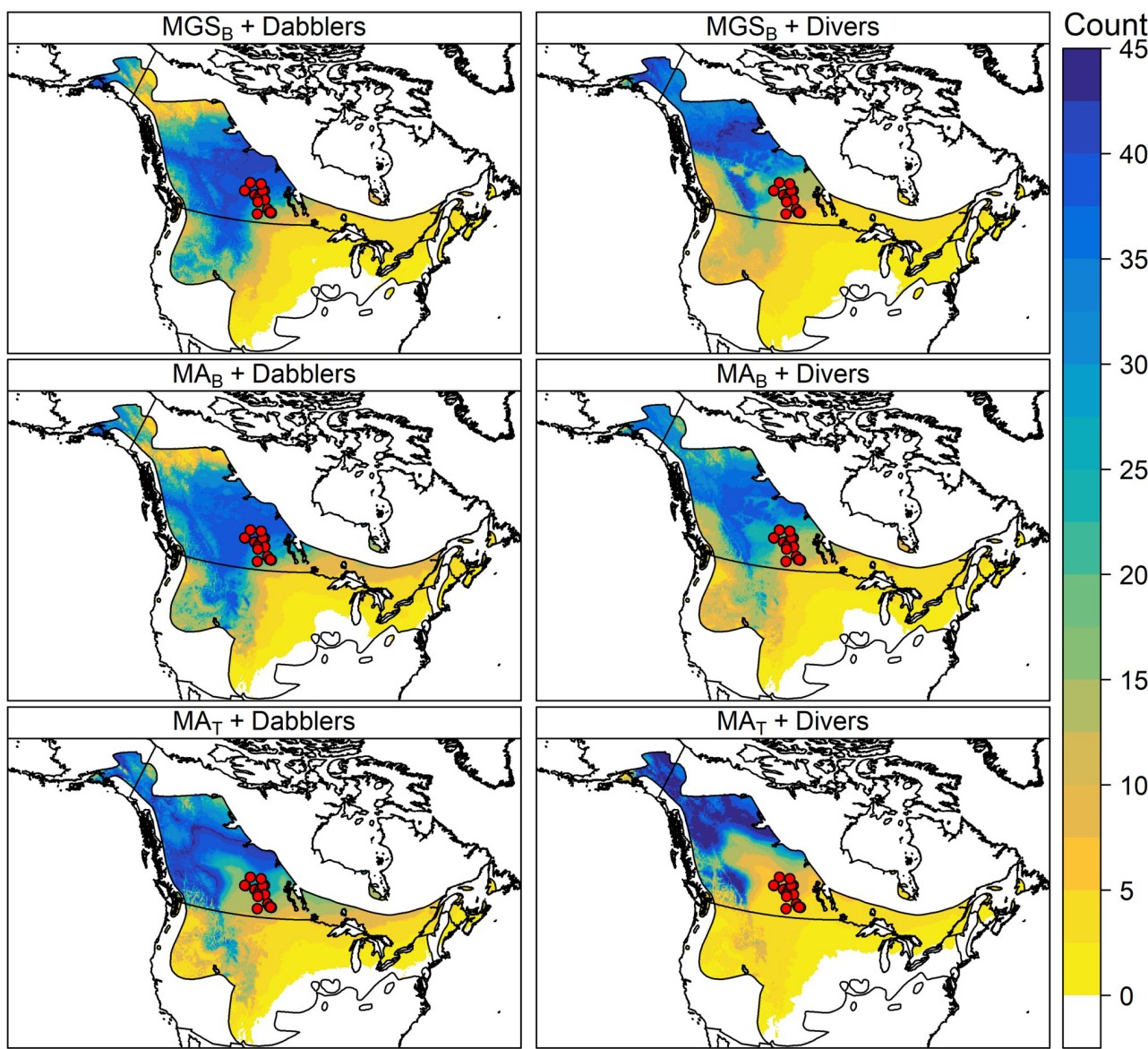

**Fig 5. Test dataset results.** Likely origins of Blue-winged Teal (*Spatula discors*) harvested in southern Saskatchewan (n = 47, 2014–2018 [9]) using different assignment methods. Panels show likely origins determined using different precipitation isoscapes (amount-weighted mean growing-season precipitation, MGS$_B$ [50]; amount-weighted mean annual precipitation, MA$_B$ [50]; MA$_T$ [56]) and calibration equations (listed in the panel strip text: isoscape + calibration equation). The colour indicates the number of individuals that were assigned to a given pixel under a 2:1 odds ratio. Harvest locations for samples are shown as red points.

Europe. As with numerous other studies, we found a strong positive relationship between $\delta^2H_f$ and $\delta^2H_p$ values, reinforcing the usefulness of using stable isotopes to determine likely origin for unknown-origin individuals. The MA and MGS isoscapes showed similar relationships with known-origin $\delta^2H_f$ values, suggesting that either MA or MGS is suitable for predicting $\delta^2H_f$ values. Finally, when we applied these different assignment methods to a test dataset, the region of most likely origin remained consistent overall, with some minor discrepancies.

Few studies have directly compared the suitability of MA and MGS methods for deriving isoscapes and relating them to consumer tissues. Bowen et al. [38] compared the relationship

between $\delta^2H_p$ and $\delta^2H_f$ for known-origin North American and European birds (based entirely on [17] for North America and [26] for Europe), using their derived MA and MGS grids, and found that neither isoscape fit significantly better (NA: MA $r^2 = 0.67$ and MGS $r^2 = 0.65$; EU: MA $r^2 = 0.85$ and MGS $r^2 = 0.86$). In this study and ours, European birds showed a slightly better fit compared to North American birds, as demonstrated by the van Dijk dataset which had the highest coefficient of determination ($r^2 = 0.65$–$0.67$) compared to the North American datasets ($r^2 = 0.27$–$0.45$). We found that the van Dijk dataset showed the lowest minimum distance measures, which could be driven by the relatively smaller area in Europe compared to North America.

Our results suggest that MA or MGS perform equally well for predicting surface water inputs into food webs for foraging waterfowl, regardless of foraging strategy, as both $\delta^2H_p$ values correlated comparably with known-origin $\delta^2H_f$ values. It is worth noting that calibration relationships using MA or MGS $\delta^2H_p$ never explained more than ~ 50–60% of the variance. These values are consistent with what has been seen in other single-species or multi-taxa studies (e.g., Accipitriformes and Falconiformes $r^2 = 0.64$ [37]; Charadriiformes, Columbiformes, Galliformes, Passeriformes, and Piciformes $r^2 = 0.54$–$0.66$ [26]; Northern long-eared bat *Myotis septentrionalis* $r^2 = 0.47$–$0.53$ [32]), although other, better fitting, examples exist (e.g., Chiroptera $r^2 = 0.72$ [24]; Coleoptera $r^2 = 0.74$–$0.78$ [76]; Odonata $r^2 = 0.75$ [28]; Passeriformes $r^2 = 0.83$ [17]; see S1 Table for further examples). The single most likely contributor to this variation for waterfowl is the mismatch between predicted food web water $\delta^2H$ based on precipitation and that manifested on the ground. Other authors have pointed to the effects of evapotranspiration in small wetlands [53] as well as differential inputs from snowmelt and complex hydrology [77], but other sources of inter- and intraspecific variation, such as diet, timing of moult, and metabolic routing [52,78], also undoubtedly contribute. This unexplained variability within these calibration models may be enough to mask these differences in fit between the MA or MGS methods, nullifying the usefulness of more specific or more general $\delta^2H_p$ measurements.

We initially expected that MA $\delta^2H_p$ would provide better integration of water isotope data compared to MGS $\delta^2H_p$ values for northern-origin individuals. The contribution of snowmelt to waterfowl feathers would be more pronounced for individuals breeding in the far north, where prolonged colder temperatures lead to more snowfall and a greater contribution of snowmelt to waterbodies. Here we would see relatively more negative $\delta^2H_f$ values due to the greater contribution of snowmelt, which for a given location should have relatively more negative $\delta^2H_p$ values compared to rain [79]. For our sampling in eastern Canada and the USA, these far northern individuals were mostly unavailable as we relied entirely on pre-existing banding operations to collect feathers, none of which were farther north than ~50 ˚N. For southern-origin waterfowl, fewer months below freezing means the MGS grid approximates the MA grid (S2 Fig), so these differences are mostly negligible. At more northern latitudes, with colder climates, we would expect a greater contribution of snowmelt [53]. What we found instead was greater variability in $\delta^2H_f$ for known-origin ducks in regions with more negative $\delta^2H_p$ values, which generally are found in the far north. In the Hebert dataset, individuals sampled at locations with lower $\delta^2H_p$ values ($< -100$ ‰), variability in $\delta^2H_f$ was greater, with more $\delta^2H_f$ values being lower than the predicted $\delta^2H_f$ values. Many individuals in this range were also removed as outliers before analysis (n = 18). From these outliers, the majority were found in the prairies (n = 15; 14 Mallard and 1 Northern Pintail) although they were not restricted to a specific species/region and did not represent the entirety of samples at any given site. This is surprising as we would expect relatively higher $\delta^2H_f$ values due to surface water becoming progressively more enriched in deuterium during evaporative processes [53,79], but this was not the case. For these outliers, especially for those in the far north (n = 1 Alaska,

n = 2 Northwest Territories), this may indicate that snowmelt was disproportionately important, but without further investigation, it is difficult to be sure. That said, we still see many individuals clustered around the predicted $\delta^2H_f$ values.

Comparing calibration equations derived from dabblers and divers, there were clear differences, but the sample size disparity between the two was significant. As such, it is difficult to validate the performance of the diving duck data. When our additional diving duck samples were added to the Clark dataset, performance decreased across all measures, including accuracy. Although accuracy decreased slightly, this is not unexpected as this equation included multiple diving duck species compared to the Clark dataset which was only Lesser Scaup. It is clear from these results that additional samples for known-origin diving ducks should be collected to build a more robust diving duck calibration dataset.

## Limitations

Propagating realistic estimates of error into isotopic assignments using isoscapes is integral to achieving realistic results useful for conservation and management [57]. Although we incorporated calibration error in our likelihood-based assignments, we did not explicitly account for isoscape interpolation error when validating the $MA_T$ grid, as this measure was unavailable. If these error estimates are available in the future, our methods can properly incorporate isoscape error into these assessments. Regardless, our current focus on choosing the best calibration algorithm remains unaffected as even without this error measurement, the assignments for the test dataset showed overall the same regions of likely origin.

We did not account for age-effects in our model, despite including adults in the samples we collected and those from the van Dijk dataset [44]. In [44], they found that accounting for age effects, while simultaneously controlling for year effects, resulted in a marginally better calibration model fit (0.61 compared to 0.71). Here, juvenile feathers showed lower $\delta^2H_f$ values compared to adults (difference = –6.8 ‰), which is consistent across other avian taxa, such as American Redstart *Setophaga ruticilla* [80], Bicknell's Thrush *Catharus bicknelli* [51], Cooper's Hawk *Accipiter cooperii* [81], and Ovenbird *Seiurus aurocapilla* [82]. This effect is likely to be driven by adult birds experiencing higher body water loss due to increased provisioning effort before moult leading to enriched $\delta^2H_f$ values [51], different feather growth rates, or dietary routing or microclimate differences between nestlings and adults [80]. In practice, this difference in $\delta^2H_f$ of adults and juveniles may not lead to a noticeable difference in the assignment. For example, using the Blue-winged Teal test dataset, if we randomly select an individual (ID = 2014_SK-01, $\delta^2H_f$ = -159.3 ‰), increase and decrease the $\delta^2H_f$ value by 6.8 ‰, and repeat the assignment procedures, we get a distance of 192 and 383 km between the centroids of the resulting binary regions and the centroid of the binary region from the original value. With the geographic scales that we are working with in most of these assignments (usually the breeding range of a species), these distances would be negligible. Regardless, for our analyses, this information was not available in the assignR database, but access to this information would improve the usefulness of these and future known-origin data.

We relied on published isoscape grids rather than year-specific, month-specific, or other custom isoscapes produced using platforms such as isoMAP [83]. This choice served two main purposes: 1) these freely-available grids are the main isoscapes already used in waterfowl calibration studies (only three publications used custom surfaces [36,72,84], other than the kriged surfaces used before 2005) and 2) these grids represent the most user-friendly source of isotope data. Further, while short-term $\delta^2H_p$ measures may be more specific, they often result in increased uncertainty due to reduced spatial coverage from sampling points [85]. At the time of this publication, the isoMAP server was unsupported and may not be available for future

studies. Overall, our intention here was to provide actionable and easily accessible recommendations that can be used by waterfowl managers and researchers.

We assembled as large a sample of known-origin waterfowl feathers as possible to maximize power for describing calibration relationships although our work could be further improved with larger sample sizes. For example, banding operations, especially those occurring in remote areas, should consider collecting feathers from local HY birds during regular banding operations. Several other studies contain known-origin duck tissues [53,86], which could be integrated with our growing database to define these relationships. Another valuable, but uncommon, source is banded known-origin HY birds submitted to the North American Waterfowl Parts Collection [87] and Species Composition Surveys [62], although using these incidental sources of feather collection comes with some necessary assumptions (e.g., harvested individuals have not opportunistically regrown feathers since banding). These surveys are excellent sources of feathers from harvested birds [9,10,15] but have been underutilized to date. All $\delta^2H_f$ data used here are available [88] for future calibration studies and help build upon the literature describing these relationships (see S1 Table). As this isotopic database grows, researchers will be able to combine and compartmentalize the data to directly derive the necessary calibration equations from their own environmental water measurements or other custom isoscape. This workflow is recommended in packages like assignR [58]. Adding to these databases allows us to not only better describe these $\delta^2H_f \sim \delta^2H_p$ relationships, in a changing world, but also to refine these into more specific (e.g., by taxa, age, diet) calibration relationships, as necessary.

## Recommendations

Waterfowl conservation and management can benefit greatly from the adoption of stable-isotope methods. These assignment techniques are not new to waterfowl applications [5–9,11–14,73,74] but have yet to be used in routinely or other than for a few specific species. In the early days of isotopic assignment, use of published calibration equations was often necessary, but we are now at the stage where users can use publicly available known-origin data to derive or supplement calibrations as needed. Whether deriving the calibration using such data or using the equations presented here (see Table 3), we recommend the use of the combined foraging-guild-specific calibration datasets (Dabblers, MGS$_B$: $\delta^2H_f$ = -69.9 + 0.7 * $\delta^2H_p$; MA$_B$: $\delta^2H_f$ = -71.7 + 0.6 * $\delta^2H_p$ and Divers, MGS$_B$: $\delta^2H_f$ = -82.6 + 0.5 * $\delta^2H_p$; MA$_B$: $\delta^2H_f$ = -78.4 + 0.5 * $\delta^2H_p$) for general applications to assign unknown-origin waterfowl in North America and Europe. For regional and species-specific studies, such as the assignment of unknown origin Mallards in Europe, the use of the more specific dataset for that species and/or region (van Dijk ~ MGS$_B$ in this case) is warranted. Although the Dabblers dataset showed lower precision and model fit compared to the individual dabbling duck datasets, we consider the Dabblers dataset to be the most conservative and realistic relationship. Similarly, for diving ducks our derived calibration equation for the combined Divers dataset offers little improvement over the Clark dataset, but we still recommend using the more general dataset including multiple diving duck species, unless the application is for Lesser Scaup specifically. While the RCWIP2 isoscape has advantages based on more advanced algorithms [56], it is computationally challenging due to computer memory requirements and currently has no associated error estimates. Although [56] lists the "40-fold" increase in resolution as an overall improvement, use of the RCWIP2 isoscape is currently limited. Ultimately, for watefowl, neither MGS$_B$ nor MA$_B$ measurements presented a markedly better relationship and the use of either grid could is justified, although precision was marginally better for MA$_B$. Refining these relationships is important, but understanding the limitations of the approach is absolutely necessary to interpret results from isotopic assignment methods.

## Supporting information

**S1 Fig. Recreation of original figure from [9].** Likely origins of Blue-winged Teal (*Spatula discors*) harvested in southern Saskatchewan (n = 47, 2014–2018 [9]) using the assignment methods from the original publication (calibration: $\delta^2H_f = -31.6 + 0.93 * \delta^2H_p$; $SD_{resid}$ = 12.8). The colour indicates the number of individuals that were assigned to a given pixel under a 2:1 odds ratio. Harvest locations for samples are shown as red points.
(TIF)

**S2 Fig. Spatial representation of differences in $\delta^2H_p$ between precipitation isoscape methods, for North America and Europe.** Each panel shows the difference (first isoscape minus second) between paired isoscapes ($MGS_B−MA_B$, $MGS_B−MA_T$, $MA_B−MA_T$): amount-weighted mean growing-season precipitation [50] ($MGS_B$) and amount-weighted mean annual precipitation ($MA_B$, [50]; $MA_T$, [56]). Blue regions represent areas where the first isoscape is much more positive than the second and yellow regions represent areas where the first isoscape is more negative than the second.
(TIF)

**S1 Table. Summary table for published calibration equations and known-origin data.** For a detailed description of the table fields, see the README.txt file.
(ZIP)

## Acknowledgments

We acknowledge the countless hours that numerous banders and volunteers contributed to capture ducks and collect feathers. Without this effort, we would not be able to achieve such a massive geographic spread in known-origin feathers. In particular, from Canada, we thank Rod Brook, Shawn Meyer, Norm North, Bruce Pollard, Jean Rodrigue, Denby Sadler, Ayden Sherritt, and Mathieu Tétreault. From the USA, we thank Donald Avers, Gary Costanzo, Drew Fowler, Min Huang Huesman, Nathaniel Huck, Benjamin Lewis, Ted Nichols, Nathan Simmons, and Jason Winiarski. We thank Robert Clark, Jacintha van Dijk, Craig Hebert, Marcel Klaassen, Shawn Meyer, Włodzimierz Meissner, Matthew Palumbo, Christian Roy, and Leonard Wassennar, for the use of their data. The Scientific Advisory Committee of the Long Point Waterfowl and Wetlands Research Program of Birds Canada provided comments to improve the paper. We thank the two anonymous reviewers for their careful reading of the manuscript and their insightful comments.

## Author Contributions

**Conceptualization:** Jackson W. Kusack, Douglas C. Tozer, Michael L. Schummer, Keith A. Hobson.

**Data curation:** Jackson W. Kusack, Kayla M. Harvey.

**Formal analysis:** Jackson W. Kusack.

**Funding acquisition:** Douglas C. Tozer, Michael L. Schummer, Keith A. Hobson.

**Investigation:** Jackson W. Kusack, Kayla M. Harvey, Keith A. Hobson.

**Supervision:** Douglas C. Tozer, Michael L. Schummer, Keith A. Hobson.

**Visualization:** Jackson W. Kusack.

**Writing – original draft:** Jackson W. Kusack, Keith A. Hobson.

**Writing – review & editing:** Jackson W. Kusack, Douglas C. Tozer, Kayla M. Harvey, Michael L. Schummer, Keith A. Hobson.

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
