## [Decision Letter · Decision Letter 0]

19 Apr 2023

PONE-D-23-08212Assigning harvested waterfowl to geographic origin using feather *δ*^2^H isoscapes: What is the best analytical approach?PLOS ONE

Dear Mr. Kusack,

Thank you for submitting your manuscript to PLOS ONE. I have now received two reviews of the manuscript and both reviewers think the manuscript makes a positive contribution to the understanding and application of hydrogen isotope analysis in bird feathers. Given the recommendations for acceptance of the manuscript with minor clarifications, I am pleased to invite you to submit a revised manuscript that takes into account the the editing suggestions of the two reviewers.

We look forward to receiving your revised manuscript.

Kind regards,

Lee W Cooper, Ph.D.

Section Editor

PLOS ONE

Journal Requirements:

3. We note that you have stated that you will provide repository information for your data at acceptance. Should your manuscript be accepted for publication, we will hold it until you provide the relevant accession numbers or DOIs necessary to access your data. If you wish to make changes to your Data Availability statement, please describe these changes in your cover letter and we will update your Data Availability statement to reflect the information you provide

Reviewers' comments:

Reviewer's Responses to Questions

**Comments to the Author**

1. Is the manuscript technically sound, and do the data support the conclusions?

Reviewer #1: Yes

Reviewer #2: Yes

2. Has the statistical analysis been performed appropriately and rigorously? 

Reviewer #1: Yes

Reviewer #2: Yes

3. Have the authors made all data underlying the findings in their manuscript fully available?

Reviewer #1: Yes

Reviewer #2: No

4. Is the manuscript presented in an intelligible fashion and written in standard English?

Reviewer #1: Yes

Reviewer #2: Yes

5. Review Comments to the Author

Reviewer #1: Analyses in this paper are well presented and the results well supported. There is some confusion in numbering of figures – apparently what had been Fig S1 in Supporting Information has been moved to the main text, without correcting the numbering of the figures between the main text and Supporting Information (see P 36). There are also some column alignment problems with Table S1. Below are some specific comments to clarify the text.

1. L 26. For the general reader, please use a phrase that explains more clearly what "amount weighted" means. Amount of what?

2. L 36-37. L 36-37. I suggest "data for individual dabbling duck species"

3. L 35-37. What about diving ducks? As they were also included in the study, the reader expects some analogous comments.

4. L 54. Replace "that" by "where'

5. L 96-97. But only if the juveniles are collected before migration. I have seen HY diving ducks that were still molting head feathers upon arrival at wintering areas. Unless you collect the juveniles before they migrate, you will still need to use only flight feathers.

6. L 206. I presume that "Fig 1" here refers to Fig S1 that is now in Supporting Information, as the caption to Fig S1 is missing from the Supporting Information. Please reconcile the numbering, placement, and captions of your figures.

7. L 218. Cornell is misspelled.

8. L 514. I suggest inserting "choice" after "This".

9. L 529. Remove the comma after "source" and reinsert after "uncommon", or else remove both commas around this phrase

10. L 540. Delete the comma after "relationships"

11. L 550. Replace “this” by “the Dabbler dataset”. I recommend avoiding use of “this” as a noun, as it is often unclear what “this” refers to.

Reviewer #2: Kusack et al. evaluate the relationship between feather and precipitation hydrogen isotope values for known-origin waterfowl. They used both previously published data from several species, as well as new data collected specifically for the present study. Their goal was to assess which isoscape (mean annual or growing season) yielded the most precise and accurate geographic assignments for different groups (dabbling vs diving) of ducks, with the goal of using this information to improve assignments of unknown-origin waterfowl to their most likely molting locations. The results indicate that mean annual and growing season isoscapes are equally suitable for application to waterfowl (regardless of foraging strategy) and that there were only small differences among the performance of the know-origin datasets. The methods used are generally reliable and the interpretations sound, though I offer a number of specific suggestions/comments below to help improve the manuscript.

My one big-picture comment is related to which calibration equation(s) listed in Table 3 the authors recommend future studies use. That isn’t entirely clear/explicit. For example, if I am studying Mallards should I use van Dijk or the combined dabblers dataset? Relatedly, if I am working on a species of waterfowl other than those presented in this manuscript, should I take the time/effort/expense in developing a calibration equation for my own species or would the dabbling or diving duck datasets be sufficient? More guidance for readers who are interested in using the results of this study for their own species/system would be helpful.

Specific comments

Lines 25-27: The wording here is unclear. I think the authors are intending to say that calibrations for most aquatic and semi-aquatic species are also currently done using amount-weighted mean growing-season d2H values, but the calibration relationship may not be not as clear for them as for terrestrial-foraging species. However, that isn’t what this sentence says. Perhaps delete everything before the first comma to fix.

Line 30: Perhaps indicate that 3 of the 4 datasets were previously published and one was collected as part of the current study.

Lines: 37-38: What dataset do the authors recommend be used for diving ducks?

Line 39: What “manner” are the authors referring to?

Line 55: What “information” are the authors referring to?

Line 57: Please add citations to back up this claim.

Line 73: “isotopic source and routing” is jargony. Consider deleting everything after the comma in this sentence.

Lines 81-82: Consider changing the order lepidopterans to a common name (butterflies and moths) to match the rest of the list.

Line 115: Why is this important?

Line 118: Please remove the second “Table”.

Line 121: What “relationship” are the authors referring to? Similarly, the relationship is “less clear” than what?

Line 124: I see >1 calibration relationship in Table 1 that isn’t based on MGS.

Line 125: Need to adjust the wording here. There are obviously many studies that have measured d2H values of surface water. There are not many studies (as I believe the authors intend to indicate) that have measured d2H values of surface water for the purpose of comparison with d2H of feathers from known-origin individuals.

Line 136: This is the first mention of foraging guilds of ducks in the paper outside of the abstract. The audience should know why you’re interested in exploring the calibration relationship for dabbling/diving ducks prior to the end of the introduction where you are stating your research questions and goals.

Lines 137-141: This may be more useful earlier on in the introduction to better introduce the importance of understanding calibration relationships for the foraging guilds of ducks.

Line 144: “underrepresented” relative to what?

Line 158: What does “largely” mean in this context?

Lines 161-162: Should “MGS-T” instead be “MA-T”?

Line 165: Do the MGS-B and MA-B isoscapes cover the period of 1960-1999? If so, perhaps say that earlier in the text to help the reader understand.

Lines 166-167: Why did you include these additional predictors in addition to the spatial correlates, and are these predictors significant to model performance?

Readme.text file: The file indicates if the data are or aren’t available for download, but it isn’t clear to me where the data that are available can be obtained. Please make the data available in association with this publication or at least indicate where readers can download the data.

Table 2: Does “n” refer to the number of birds or feathers?

Line 218: Please change “Cornel” to “Cornell”

Lines 237-239: Good, but were the sample standards used? And were the same d2H values for those standards used, given that their values have changed, as indicated in Soto et al. 2017? If not, reference 63 (Magozzi et al.) provides an approach to deal with this issue.

Lines 265-267: It isn’t clear to me why these particular locations are specified here if they are already outside the bounds of North America and Europe.

Line 279: Please include a citation for the isocat R package.

Lines 280-281 and 285-290: Why was the odds ratio approach used? Campbell et al. 2020 showed that when known-origin individuals are available (or even when they aren’t), there are other approaches for assignment that have better accuracy/precision than the odds ratio.

Line 281: “uniform” in what way?

Lines 281-283: Were the samples that the grouped datasets contained from outside NA and EU, not known-origin? Is that why you’re excluding the other regions? Also, are the grouped datasets referring to the ‘Dabblers’ and ‘Divers’ dataset?

Line 290: It isn’t clear to me why a 2:1 odds ratio is “conservative”

Line 297: You used the function ‘distance’ as opposed to what function from [56]?

Line 306: Please change “subset” to the past tense (“subsetted”).

Line 322: Odd wording. Possibly reword “because we identified this site..” to something like “due to the site’s location on a cranberry farm”.

Line 338: Model fit appears worse, not improved, for the combined diver dataset compared to the Clark dataset (as lines 481-482 also say). Please clarify.

Table 3: Please also report sample sizes. Also, please define “MA-T”.

Lines 431-433: It isn’t immediately clear to me why a larger range of d2H values would necessarily lead to a lower fit.

Lines 487-489: Could the error associated with the growing season grid be applied to the MA-T grid? Seems like that would be better than not accounting for interpolation error?

Lines 519-522: These last two sentences are tagged on to this paragraph and don’t really make sense.

Line 535: I cannot find this appendix.

Lines 545-547: Please be specific about which equation(s) in Table 3 is being recommended.

Lines 552-552: which is/are “the most recent calibration equations” being recommended?

Line 558: Unless I’m reading Table 3 incorrectly, I believe -71.94 should be -71.75. Or perhaps the table is incorrect.

Figure S2: : I don’t believe Figure S2 is referenced anywhere in the manuscript. Also, what does “MA-T” indicate?

6. PLOS authors have the option to publish the peer review history of their article (what does this mean?). If published, this will include your full peer review and any attached files.

Reviewer #1: No

Reviewer #2: No

---

## [Author Response · Author response to Decision Letter 0]

3 Jun 2023

Reviewer #1: Analyses in this paper are well presented and the results well supported. There is some confusion in numbering of figures – apparently what had been Fig S1 in Supporting Information has been moved to the main text, without correcting the numbering of the figures between the main text and Supporting Information (see P 36). There are also some column alignment problems with Table S1. Below are some specific comments to clarify the text.

Response: Thank you for the comments! The column alignment in S1 Table has been made consistent (left justified). All other comments are addressed in the response to reviewers document. We also address the issues with figure numbers there. 

Reviewer #2: Kusack et al. evaluate the relationship between feather and precipitation hydrogen isotope values for known-origin waterfowl. They used both previously published data from several species, as well as new data collected specifically for the present study. Their goal was to assess which isoscape (mean annual or growing season) yielded the most precise and accurate geographic assignments for different groups (dabbling vs diving) of ducks, with the goal of using this information to improve assignments of unknown-origin waterfowl to their most likely molting locations. The results indicate that mean annual and growing season isoscapes are equally suitable for application to waterfowl (regardless of foraging strategy) and that there were only small differences among the performance of the know-origin datasets. The methods used are generally reliable and the interpretations sound, though I offer a number of specific suggestions/comments below to help improve the manuscript.

My one big-picture comment is related to which calibration equation(s) listed in Table 3 the authors recommend future studies use. That isn’t entirely clear/explicit. For example, if I am studying Mallards should I use van Dijk or the combined dabblers dataset? Relatedly, if I am working on a species of waterfowl other than those presented in this manuscript, should I take the time/effort/expense in developing a calibration equation for my own species or would the dabbling or diving duck datasets be sufficient? More guidance for readers who are interested in using the results of this study for their own species/system would be helpful.

Response: Thank you for your detailed review and for the constructive comments. We’ve tried to simplify our recommendations and provide more detailed responses in the response to reviewers document.

---

## [Editor Report · Decision Letter 1]

15 Jun 2023

PONE-D-23-08212R1Assigning harvested waterfowl to geographic origin using feather *δ*^2^H isoscapes: What is the best analytical approach?

PLOS ONE

Dear Dr. Kusack,

Thank you for submitting your manuscript to PLOS ONE. After careful consideration, we feel that it has merit but does not fully meet PLOS ONE’s publication criteria as it currently stands. Therefore, we invite you to submit a revised version of the manuscript that addresses the points raised during the review process.

Thank you for submitting your revised manuscript to PLOS ONE. I appreciate the time and effort you and your co-authors have made in revising the manuscript in response to the two reviews of the original manuscript. I have read through the revised manuscript and I think you have adequately addressed the reviewer comments and recommendations and it is not necessary to send the manuscript back to the reviewers. Therefore, I think the manuscript is acceptable in its current form and will be useful in the future for understanding waterfowl origins based upon the hydrogen isotope approach. I did see one typographical error at line 327 in the clean version of the paper, "a" appears twice before cranberry farm. Also, you don't have to, but I thought the reviews were constructive and helpful in improving the paper, so I'd ask that you consider thanking the two anonymous reviewers for their help in improving the manuscript in the acknowledgements section. If you can address those two minor editorial concerns, I will be pleased to recommend to the editorial office that the manuscript be accepted for publication.

We look forward to receiving your revised manuscript.

Kind regards,

Lee W Cooper, Ph.D.

Section Editor

PLOS ONE

---

## [Editor Report · Decision Letter 2]

22 Jun 2023

Assigning harvested waterfowl to geographic origin using feather *δ*^2^H isoscapes: What is the best analytical approach?

PONE-D-23-08212R2

Dear Dr. Kusack,

Thank you for making the last few changes I suggested to your manuscript. I am pleased to recommend to the editorial office that your manuscript has been judged scientifically suitable for publication and will be formally accepted for publication once it meets all outstanding technical requirements.

Kind regards,

Lee W Cooper, Ph.D.

Section Editor

PLOS ONE

---

## [Editor Report · Acceptance letter]

30 Jun 2023

PONE-D-23-08212R2 

Assigning harvested waterfowl to geographic origin using feather *δ*^2^H isoscapes: What is the best analytical approach? 

Dear Dr. Kusack:

I'm pleased to inform you that your manuscript has been deemed suitable for publication in PLOS ONE. Congratulations! Your manuscript is now with our production department. 

Kind regards, 

on behalf of

Dr. Lee W Cooper 

Section Editor

PLOS ONE